# Tumour-on-chip microfluidic platform for assessment of drug pharmacokinetics and treatment response

Tudor Petreus[1], Elaine Cadogan [1], Gareth Hughes[1], Aaron Smith [2], Venkatesh Pilla Reddy[2], Alan Lau[1], Mark James O'Connor[1], Susan Critchlow[1], Marianne Ashford [3] & Lenka Oplustil O'Connor [4✉]

Microphysiological in vitro systems are platforms for preclinical evaluation of drug effects and significant advances have been made in recent years. However, existing microfluidic devices are not yet able to deliver compounds to cell models in a way that reproduces the real physiological drug exposure. Here, we introduce a novel tumour-on-chip microfluidic system that mimics the pharmacokinetic profile of compounds on 3D tumour spheroids to evaluate their response to the treatments. We used this platform to test the response of SW620 colorectal cancer spheroids to irinotecan (SN38) alone and in combination with the ATM inhibitor AZD0156, using concentrations mimicking mouse plasma exposure profiles of both agents. We explored spheroid volume and viability as a measure of cancer cells response and changes in mechanistically relevant pharmacodynamic biomarkers ($\gamma$H2AX, cleaved-caspase 3 and Ki67). We demonstrate here that our microfluidic tumour-on-chip platform can successfully predict the efficacy from in vivo studies and therefore represents an innovative tool to guide drug dose and schedules for optimal efficacy and pharmacodynamic assessment, while reducing the need for animal studies.

[1] Bioscience, Oncology R&D, AstraZeneca, Cambridge, UK. [2] DMPK, Oncology R&D, AstraZeneca, Cambridge, UK. [3] Advanced Drug Delivery, Pharmaceutical Sciences, R&D, Macclesfield, UK. [4] Translational Medicine, Oncology R&D, AstraZeneca, Cambridge, UK.
✉email: lenka.oplustiloconnor@astrazeneca.com

A useful in vitro platform for drug discovery and development should predict response to treatment in a short period of time, be reliable and deliver results that have translational relevance. Microphysiological in vitro systems are probably the most advanced platforms for preclinical evaluation of drugs effects to date. However, platforms available on the market have their limitations such as low throughput, high cost, and more importantly inability to reproduce physiological exposures of drugs. Hence, there is a need to improve these platforms and develop better predictive systems that are able to mimic drugs in vivo pharmacokinetic (PK) profile.

Tumoroid/organoid-on-chip technology combines microfluidics and a three-dimensional (3D) cell compartment[1]. The main purpose of such systems is to maintain the biological complexity of the cell models (mostly 3D cultures, including microenvironment or vascularisation)[2–5] and combine it with the ability to mimic drug assessment, similar to animal models. Most of the existing models explore tumour progression, invasion, metastasis or angiogenesis thus recapitulating human cancer pathophysiology[6]. However, the more complex they are, the lower the throughput and challenging they become. Microfluidic technology allows continuous drug perfusion at a constant flow rate[7] through a microfluidic chip handling 2D or nude, or encapsulated 3D cultures[8,9]. One of the requirements of a microfluidic system is to assess as many potential drug candidates at different concentration ranges at the same time. Some devices are addressing this problem using multiple drug gradient generators and parallel cell culture chambers[10–12]. However, different compound concentrations are only being delivered individually to a certain number of replicate wells with 2D or 3D culture models without exposing the cell construct to varying drug concentrations typical of a plasma PK profile, that is, representative of the in vivo situation. More recent advances include PK-PD platforms using custom made PDMS chip models for 2D and 3D cultures with syringe pump controlled flow rates[13,14].

Here, we introduce a microfluidic platform able to mimic drug PK profiles and treatment schedules and to evaluate drug response on complex microenvironment tumour spheroids. This platform is able to sequentially deliver specific drug concentrations at a constant flow rate mimicking up to eight concentration points on the PK profile for a specific drug. Using one disposable reservoir per concentration point, the system can mimic individual PK profiles for a single drug or combination of two compounds at a time. Furthermore, various treatment schedules can also be explored to further guide the design of in vivo or clinical studies, thus preventing long and expensive dose scheduling clinical studies. This platform includes a single channel microfluidic Ibidi chip (Ibidi GmbH, Germany), that we adapted to handle eight tumour spheroids encapsulated in Matrigel droplets. Here, we use our novel setup to reverse-translate the in vivo response to a topoisomerase-I (TOP1) inhibitor, irinotecan (active metabolite SN38) and its combination with an oral inhibitor of ATM kinase (AZD0156) on encapsulated colorectal cancer cell line spheroids (SW620). Anticancer drug response was evaluated by the changes in spheroid volume, cellular metabolic activity and quantification of biomarkers of DNA damage, cell death and proliferation. Using the same cancer model, we explored differences between response to the treatment in static (2D plate format) versus microfluidic 3D setup and between a single agent SN38 and various combination schedules with AZD0156. Finally, by comparing the outcome from in vivo studies and in vitro microfluidic platform experiments, we were able to demonstrate the predictive ability of our novel system for anticancer drug assessment.

## Results

**Design of SW620 spheroid-on-chip and the microfluidic platform**. The current microfluidic platform (Fig. 1) consists of a pump-driven system that delivers compounds on Matrigel-encapsulated spheroids in a concentration and time dependent manner, simulating pharmacokinetic profiling that occur in vivo.

Matrigel-embedded spheroids are considered attractive in vitro tools to assess response to drugs, as the Matrigel network provides a structural support and is permeable to nutrients, growth factors and small molecules (drugs, antibodies, fluorophores)[15,16]. The microfluidic platform we describe here uses SW620 spheroids, grown in a 96-well ultra-low adherence plate and then transferred in Matrigel hanging drops on a single-channel Ibidi microfluidic chip. Following Matrigel encapsulation, spheroids were exposed to a constant flow of 20 μL/min, each flow path delivering SN38 or SN38 + AZD0156 into the Ibidi chips to mimic in vivo PK profiles, and also using various treatment schedules (monotherapy, intermittent/continuous combination therapy). Our setup used two-chained chips for each treatment condition, one for viability endpoint assay and a second for pharmacodynamic biomarkers evaluation by immunofluorescence. Spheroid recovery was performed by removal of the chip's bottom polymer film at specific time points.

**SW620 tumour 3D spheroid cell response is different in static compared to microfluidic conditions**. Experiments in static plate conditions cannot effectively mimic the PK profile on the spheroids, nor can they easily be explored for treatment scheduling.

SN38 and AZD0156 plasma concentrations in mouse were taken and measured hourly for 96 h, using LC MS. In vitro experimental setup used eight concentration points translated from the 24 h plasma concentrations in mouse. Each concentration used for a specific amount of time for in vitro microfluidic setup represented an average of the corresponding time interval from observed and modelled in vivo PK data. Closely related PK profiles between mouse plasma and in vitro concentrations are illustrated in Table 1.

The concentration of a drug applied to a static set up over time is very different from the in vivo situation and the mouse PK profiles for both SN38 and AZD0156 (Fig. 2a). In addition, under static conditions in the plate format, 3D nude spheroids do not allow washing and media replacement steps due to the high risk of damage to the spheroids. In contrast, the Matrigel encapsulated spheroids in the Ibidi chip can resist the medium flow rate of 20 μL/min for up to 14 days. For flow rates greater that 20 μL/min but not exceeding 70 μL/min, the encapsulated spheroids were able to withstand the culture media flow for at least 3 days.

As a first step to compare the response of SW620 cells treated in the 3D chip microfluidic set up versus 3D plate, we explored differences in tumour spheroid size and viability following 6 days of treatment with SN38 and combination with AZD0156 using exposures as shown in Fig. 2a, b. AZD1056 monotherapy was not further explored as there was no spheroid size, morphology or viability effect on SW620 spheroids in static 2D or 3D (Supplementary Fig. 1). Tumour spheroid size at day 6 for both static 3D and the microfluidic setup was expressed as fold increase in size over day 1 values and showed that a SW620 spheroid size reduction was indicative of potentiation activity of ATM inhibitor (AZD0156) over SN38 alone, only in the microfluidic Ibidi chip but not in static experimental conditions (Fig. 2c). The untreated spheroid size at day 6 was slightly higher in the static setup possibly as these nude spheroids had no mechanical constraints (as in Matrigel-embedded spheroids, used in the microfluidic setup).

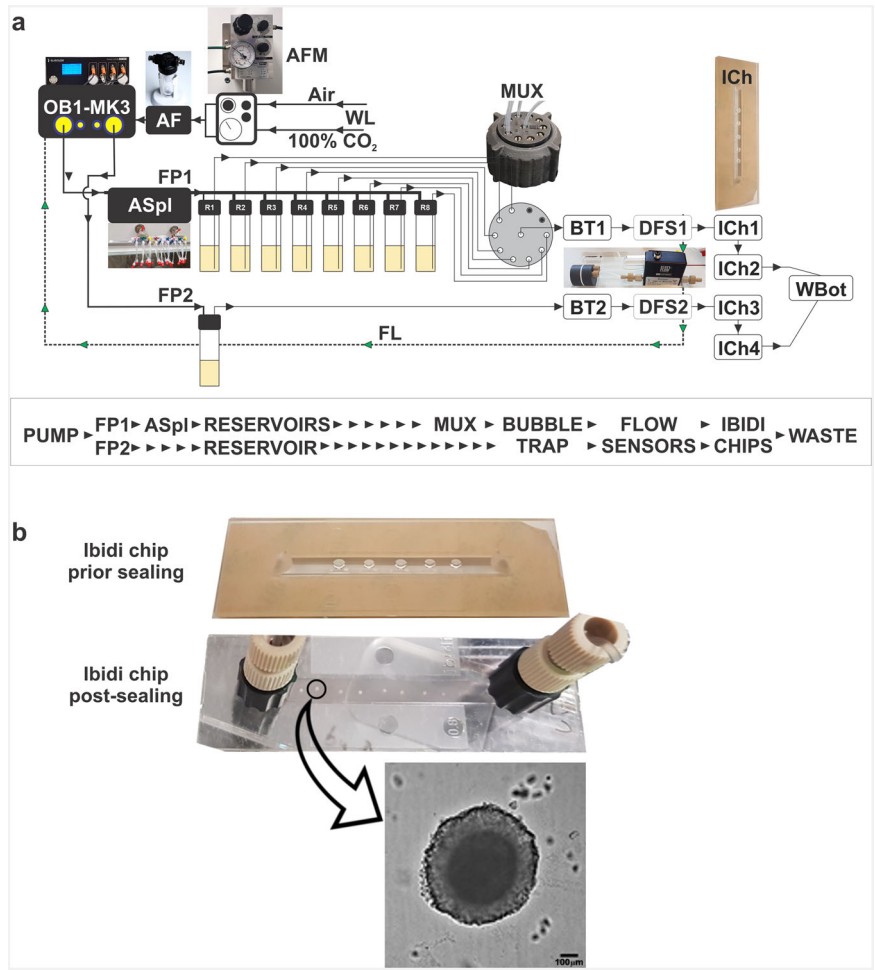

**Fig. 1 Microfluidic platform design. a** Components of the microfludic set up. Air and $CO_2$ from the wall lines (WL) are mixed in airflow mixer (AFM) and passed through air filter (AF) before reaching the pump (OB1-MK3). Flow path 1 (FP1) with air splitter (ASpl) shows eight outlets connected to the reservoirs with specific drug concentrations. Time and flow rate of the reservoir content are regulated by the distributor (MUX) and piezo pump (OB1-MK3). The FP1 continues to a bubble trapper (BT) and a digital flow sensor (DFS) into 2 chained Ibidi chips (ICh 1,2) and finaly to the waste bottle (WBot). Flow path 2 (FP2) for mock treated spheroids is represented by single reservoir and the two chained Ibidi chips (ICh 3,4). A feedback loop (FL) connects each digital sensor (DFS) to the OB1-MK3 pump to maintain desired flow rate. **b** Photograph of the Ibidi Luer chip with 8 Matrigel encapsulated spheroids, and brighfield microscopic image of SW620 spheroid after 7 days of continuous media flow. Scalebar = 100 µm.

From the same experimental setup, we explored SW620 spheroid viability at 6 days for static and microfluidic setup (Fig. 2d). We observed that continuous exposure to SN38 led to greater reduction in spheroid viability (~60%) compared to the effect caused by SN38 dosed to mimic in vivo PK profile (~40% reduction in viability). Furthermore, the SN38 + AZD0156 combination in a microfluidic setup reduced viability by 1.4-fold, compared to 2.2-fold in static setup (Fig. 2d), suggesting that the static exposure to a higher concentration of drugs may exaggerate the treatment effect. This experiment demonstrated that exposure of the 3D tumour cells to the drugs in more physiological-like setting leads to a different response than at static exposure conditions, leading us to further explore whether this novel in vitro system could indeed better predict in vivo experiments.

**Tumour-on-chip microfluidic platform can be used to explore efficacy and biomarker response to guide optimal design of treatment schedules.** Next, we assessed the response of Matrigel encapsulated SW620 spheroids in our microfluidic set up to SN38 monotherapy and in combination with AZD0156 using various treatment schedules (Table 2). The tested schedules were

designed to recapitulate mouse plasma PK profiles from similar schedules used in in vivo efficacy studies (Table 1) with a desire to reverse translate in vivo data and to evaluate the predictive ability of our microfluidic chip setup. SW620 spheroids response to treatment was assessed after 7 days. We investigated changes in the spheroid volume, as well as viability using the CellTiterGlo assay. To evaluate whether our system could also be used to monitor treatment impact on pharmacodynamic biomarkers, we collected individual spheroids at the end of the treatment period and performed immunofluorescence analysis for γH2AX (DNA double strand break marker), cleaved caspase 3 (apoptosis) and Ki67 (proliferation marker), as detailed in "Methods" section and Supplementary Fig. 2. Each treatment condition was repeated at least three times and measurements for each condition were pooled from at least six spheroid replicates.

Spheroids were imaged on day 1 and 7 (Fig. 3a) and their volume measured on day 7 following indicated treatment conditions (Table 1). A treatment effect at day 7 was assessed by calculating the volume of the spheroids following treatment as a percentage of the average volume ($n = 7$) of untreated spheroids (Fig. 3b).

**Table 1 In vivo free drug plasma concentrations (determined by LC-mass spectrometry and population-based PK model) and mimicked concentrations in the microfluidic setup.**

| In vivo drug free concentrations (nM) | | | In vitro microfluidic drug concentrations | | | |
|---|---|---|---|---|---|---|
| Time (h) | AZD0156 (10 mg/kg) | SN38 (Irinotecan 50 mg/kg) | AZD0156 (nM) | SN38 (nM) | Time (h) | Reservoir |
| 1 | 187.8 | 5.5 | 192 | 5.5 | 1 | 2 |
| 2 | 192.1 | 4.7 | 150 | 4.3 | 3 | 3 |
| 3 | 173.8 | 4.4 | | | | |
| 4 | 154.5 | 4.1 | | | | |
| 5 | 137.1 | 3.6 | 110 | 3.4 | 2 | 4 |
| 6 | 121.5 | 3.1 | | | | |
| 7 | 107.8 | 2.8 | 75 | 2.3 | 3 | 5 |
| 8 | 95.6 | 2.4 | | | | |
| 9 | 84.7 | 2.1 | | | | |
| 10 | 75.2 | 1.9 | 60 | 1.7 | 3 | 6 |
| 11 | 66.7 | 1.7 | | | | |
| 12 | 59.1 | 1.6 | | | | |
| 13 | 52.7 | 1.5 | 45 | 1.3 | 4 | 7 |
| 14 | 46.6 | 1.4 | | | | |
| 15 | 41.5 | 1.2 | | | | |
| 16 | 36.7 | 1.1 | | | | |
| 17 | 32.6 | 1.1 | 20 | 0.6 | 8 | 8 |
| 18 | 28.8 | 0.9 | | | | |
| 19 | 25.7 | 0.9 | | | | |
| 20 | 22.7 | 0.8 | | | | |
| 21 | 20.2 | 0.7 | | | | |
| 22 | 17.8 | 0.7 | | | | |
| 23 | 15.9 | 0.6 | | | | |
| 24 | 14.1 | 0.6 | | | | |

SN38 monotherapy led to a reduction in average volume size to 55% that of control. Concurrent combination treatment with SN38 (dosed on day 1) and AZD0156 (dosed on days 1–7) was more potent and caused spheroid volume reduction to 17% of control. We also explored whether the duration of AZD0156 dosing or a gap between SN38 and AZD0156 affected the combination efficacy for these particular agents. Treatment with SN38 (dosed on day 1) plus AZD0156 (dosed on day 1 only) led to a small drop in efficacy compared to a continuous 7-day treatment with AZD0156 as shown by average relative spheroid volume of 22% (1-day schedule) versus 17% (7-day schedule). Introducing a dosing gap between the SN38 and initiation of AZD0156 treatment proved to have a more significant consequence on the efficacy benefit. While a 24 h delay of AZD0156 dosing was still able to cause significant spheroid growth inhibition (25% of control volume), with a 72 h gap we measured spheroids volume was only 40% of control (Fig. 3b). Moreover, a similar outcome was observed using spheroid viability as a readout (Fig. 3c), providing additional evidence on the different impact of various treatment schedules.

SN38 is a TOP1 inhibitor that that prevents relegation of the DNA strand by binding to TOP1–DNA complex and causes DNA double strand breaks (DSBs). Those DNA DSB lesions trigger activation of the ATM kinase for effective repair. Inhibition of ATM and consequent accumulation of unrepaired breaks leading to increased cell death provides the rationale for developing this combination for cancer treatment. AZD0156 is a potent and selective inhibitor of ATM kinase, shown to provide robust efficacy against tumour models when combined with DNA DSB inducing agents[17]. Here, we tested whether different schedules lead to different activation of pharmacodynamic markers that are associated with DNA DSB damage, apoptosis and cell proliferation. The γ-H2AX associated foci accumulate in cells exposed to DBS inducing agents[18,19]. These foci can be detected by immunostaining to determine their, size and

morphology[19]. However, spheroid imaging using a 20x objective on a CV7000 confocal microscope cannot accurately explore foci morphology, so we introduced customised ImageJ macros (described in "Methods" section, Supplementary Fig. 2 and Supplementary Method 1) to quantitate positive cells within a spheroid. The same method was applied to quantitate CC3-positive (apoptotic) cells, while normalisation by spheroid area was not required for Ki67, as this biomarker is present in all proliferating cells.

We collected and imaged individual spheroids from the chip at the end of the study (day 7), following an on-chip fixing step in formaldehyde and immunofluorescence staining for γH2AX in all treatment groups (Fig. 4a, b) (Supplementary Figs. 2 and 3). We did not observe a significant change in γH2AX in spheroids treated with SN38 monotherapy compared to the control group, suggesting that the effect of single dose of SN38 on DNA damage is likely resolved by day 7. However, even a single dose of AZD0156 at concentration mimicking the mouse plasma exposure profile from a 10 mg/kg dose when combined with SN38 resulted in 68% increase in γH2AX, which was still detectable at day 7 compare to the untreated control. This was further accentuated by increasing the number of AZD0156 doses with maximum effect over 400% increase observed with 7 days of continuous exposure to ATM inhibitor after SN38 dose (Fig. 4b). Greater increase in DNA damage also resulted in potentiation of apoptosis in these cells as demonstrated by 500% increase in cleaved caspase 3 (Fig. 4c) (Supplementary Fig. 4), and it is consistent with the observed effect on spheroids size/viability (Fig. 3b, c). Proliferation of SN38 treated spheroids was about 20% decreased compared to the control cells with significantly more (50–60%) decrease in all the combination treated groups. The most obvious effect on reduction (64%) in cell proliferation was observed in the 7-day continuous treatment schedule (Fig. 4d) (Supplementary Fig. 5). Our data therefore suggest that our microfluidic system not only detects the potentiation of TOP1

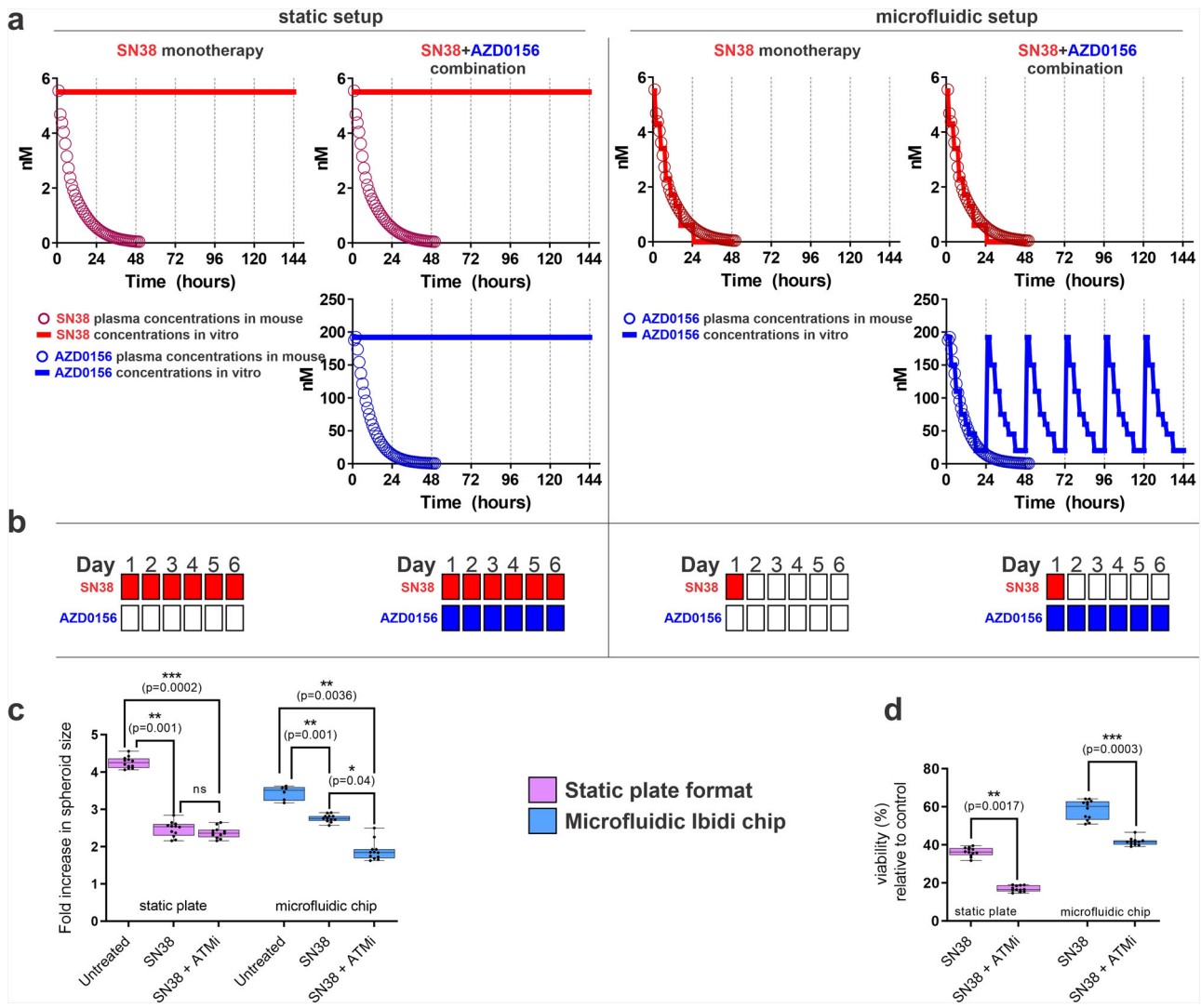

**Fig. 2 Exposure profile of SN38 and ATM inhibitor (AZD0156) in static plate format versus microfluidic Ibidi chip and consequence of different exposures on tumour cells response. a, b** In static plate format, SW620 tumour spheroids were exposed to a fixed dose corresponding to the maximum mouse plasma concentration achieved with 50 mg/kg dose of irinotecan (SN38 Cmax 5.5 nM), and 10 mg/kg AZD0156 (Cmax 192 nM) continuously for 6 days. In microfluidic Ibidi chip, tumour spheroids were exposed to eight different concentrations in a gradient fashion mimicking the in vivo pharmacokinetic profile and treatment schedule for both drugs. **c** Spheroids were imaged in situ (plate/chip) at the end of the treatment and their volumes calculated from average radius deducted from min/max Feret. **d** Spheroid viability was measured at the end of the treatment by CellTiterGlo assay (**d**). $N = 6$ per condition from two independent experiments. Box-plots show median (centre line); box limits are 25th to 75th percentile; whiskers represent min and max values. Statistical analysis was performed using 1-way ANOVA.

**Table 2 Treatment schedules for SN38 and AZD0156 assessed in the microfluidic setup, on Matrigel-encapsulated SW620 spheroids.**

| | | 1 Cycle = 7 Days | | | | | | | Treatment type and schedule |
|---|---|---|---|---|---|---|---|---|---|
| | Drug | 1 | 2 | 3 | 4 | 5 | 6 | 7 | |
| SN38 ◆ | | ◆ | | | | | | | Monotherapy |
| SN38 ◆ | AZD0156 ■1/7 | ◆■ | | | | | | | Combination (1◆ + 1/7■) |
| SN38 ◆ | AZD0156 ■7/7 | ◆■ | ■ | ■ | ■ | ■ | ■ | ■ | Combination (1◆ + 7/7■) |
| SN38 ◆ | AZD0156 ■3/7 (24 h gap) | ◆ | ■ | ■ | | | | | Combination (1◆ + 2,3,4/7■) |
| SN38 ◆ | AZD0156 ■3/7 (72 h gap) | ◆ | | | ■ | ■ | ■ | | Combination (1◆ + 4,5,6/7■) |

inhibitor induced DNA damage and apoptosis by the ATM inhibitor AZD0156, but also the impact of the duration of AZD0156 dosing after SN38 treatment, as well as the gap between the two agents.

**Microfluidic tumour-on-chip platform can predict efficacy response in vivo.** To assess the translational potential of the microfluidic platform, we designed an in vivo experiment using SW620 xenograft to assess the efficacy response to irinotecan

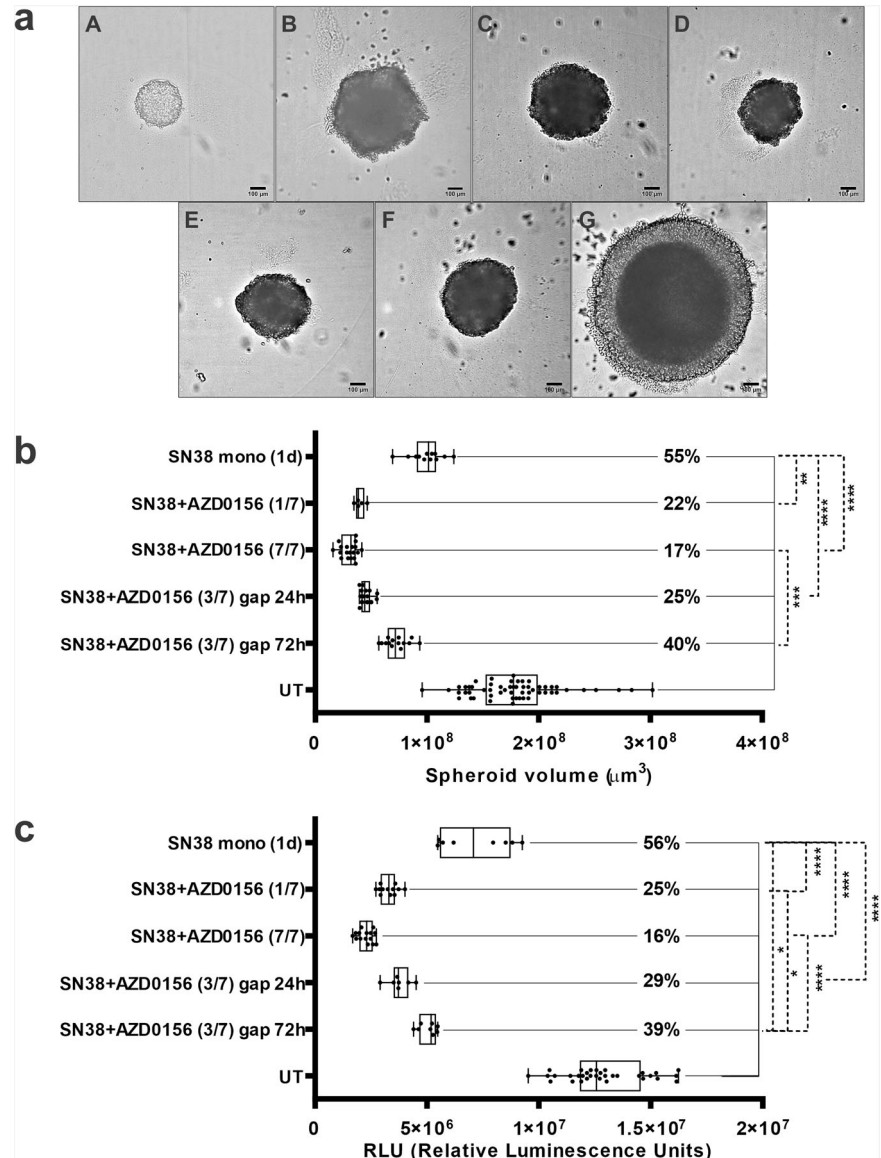

**Fig. 3 SW620 tumour spheroids response at day 7 following various schedules of SN38 and AZD0156 delivered in microfluidic Ibidi chip. a** Brightfield microscope images of SW620 tumour spheroids at day 0 and 7 days following different treatment schedules with SN38 and AZD0156: *A* Untreated day 0; *B* SN38 monotherapy; *C* SN38 + AZD0156 1/7; *D* SN38 + AZD0156 7/7; *E* SN38 + AZD0156 3/7 (24 h gap); *F* SN38 + AZD0156 3/7 (72 h gap); *G* Untreated day 7; **b** Spheroids were imaged in situ in the chip at day 7 and their volumes calculated from average radius deducted from min/max Feret. **c** Spheroids were recovered from the chip at day 7 and their viability measured by CellTiterGlo assay. $n \geq 5$ per condition from two independent experiments. Box-plots show median (centre line); box limits are 25th to 75th percentile; whiskers represent min and max values. Statistical analysis using 1-way ANOVA, Tukey's multiple comparisons, CI = 95%, ****$p < 0.0001$; ***$p < 0.001$; **$p < 0.01$; *$p < 0.1$.

(50 mg/kg) and its combination with AZD0156 (10 mg/kg). We tested different weekly in vivo dosing schedules, previously evaluated in SW620 spheroids on the chip (Fig. 5a) using matched exposure profiles to those previously observed in mouse plasma. (Fig. 2b and Table 1).

Tumour xenograft volumes (in vivo) and tumour spheroids volumes (in vitro microfluidic setup) for irinotecan, as well as irinotecan plus AZD0156 dosed with a 24 or 72 h gap, were compared at day 7. The concurrent combination schedule was not tolerated in mice, therefore in vivo efficacy data could not be generated for this. For tumour xenografts, tumour volumes were measured weekly twice for 38 days for all treatments groups, expect vehicle control, which had to be terminated on day 15 due to the tumour volume limit being reached.

There were no significant differences observed in tumour size amongst the treatment groups at day 7, all reaching approximately 80% of the control group size. However, at day 15 the combination with the 24 h gap showed greater efficacy (39% volume of control) than irinotecan alone (54% volume of control) or a combination schedule using a 72 h gap between irinotecan and AZD0156 (59% volume of control). Even more significant separation of the tumour growth curves became apparent at day 35, with combination using the 24 h gapped schedule providing the best response, followed by a combination treatment using a 72 h gap, which was still more potent than irinotecan monotherapy (Fig. 5b).

Tumour xenograft versus tumour spheroid percent growth inhibition (% GI) for the treated samples compared to untreated controls followed the same trend at 7 days for in vitro microfluidic chip setup (52% GI) and at 15 days for the in vivo experiment (53% GI) (Supplementary Fig. 6). In combination with 24 h gap treatment schedule, we observed %

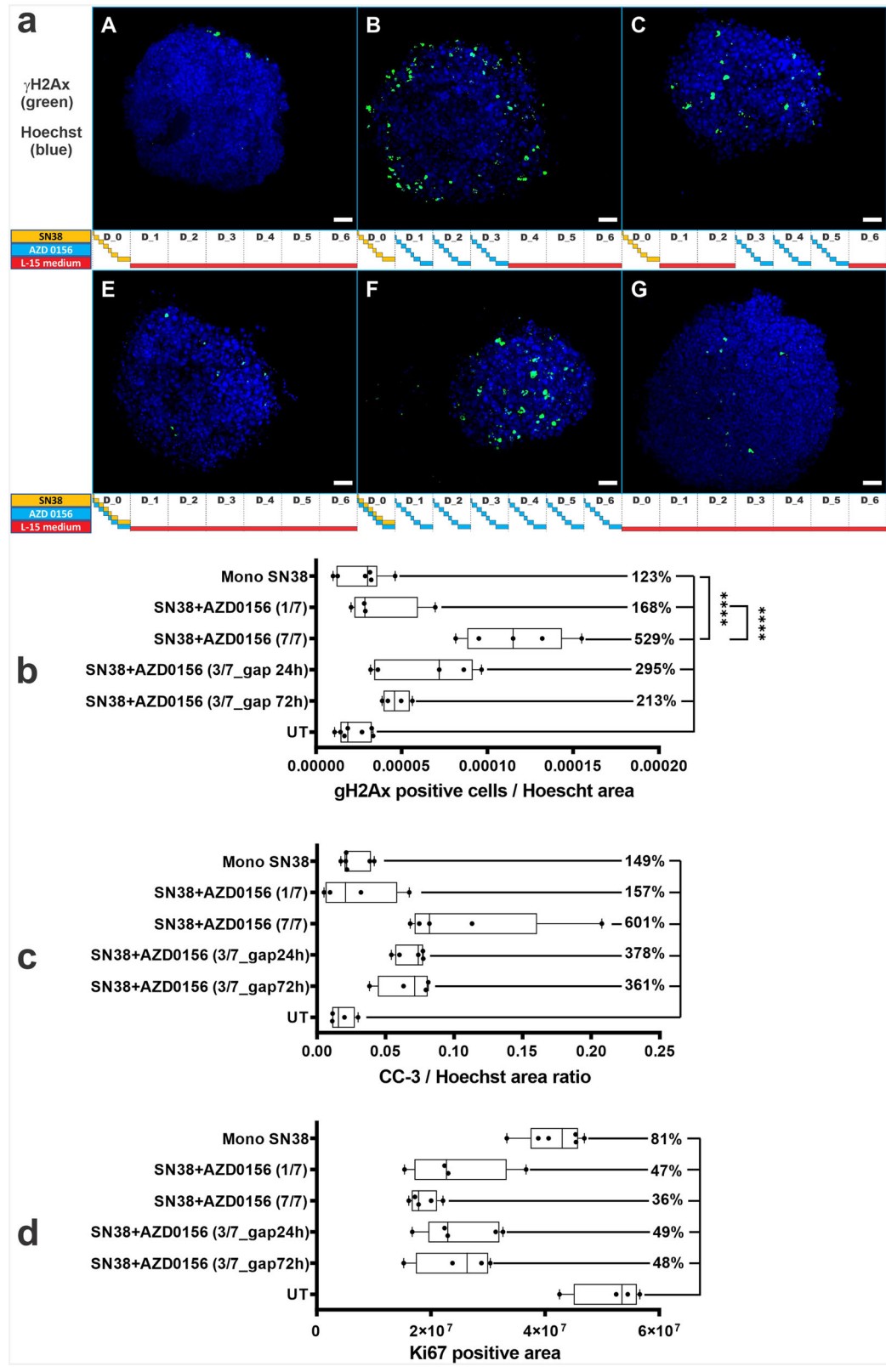

GI of 69% (at 15 days, in vivo) and 88% (at 7 days, in vitro microfluidic). Combination with 72 h gap showed 46% GI (at 15 days, in vivo) and 70% GI (at 7 days, in vitro microfluidic chip). The differences in % GI between 24 h and 72 h combination schedules were 23% (in vivo) and 18% (in vitro, microfluidic chip).

## Discussion

The aim of this work was to design a pump-driven microfluidic platform capable of mimicking in vivo anti-cancer drug PK profiles on in vitro tumour spheroids, thus demonstrating the utility of this novel platform for predicting drug PK, PD and efficacy. Nowadays, tumour-on-chip models are combining 3D spheroid culture

**Fig. 4 Immunofluorescence analysis of pharmacodynamic biomarkers in SW620 tumour spheroids following different treatment schedules in microfluidic Ibidi chip.** Spheroids were treated in the chip with SN38 and AZD0156 at different schedules using the microfluidic setup. Spheroids were recovered from the chip at day 7 and DNA double strand break damage was assessed via the presence of γH2AX. Scalebar = 100 μm. **a** Representative images of nuclei (blue/Hoechst 33342) and γH2AX (green), **b** quantification of γH2AX positive cells. (A-SN38, B-SN38 + AZD0156 3/7 with 24 h gap, C-SN38 + AZD0156 3/7 with 72 h gap, D-SN38 + AZD0156 1/7, E-SN38 + AZD0156 7/7, F-control). Cleaved caspase 3 (CC-3) was used to quantified apoptotic cell death (**c**), and Ki67 to measure the effect on proliferation (**d**). N ≥ 5 per condition from two independent experiments. Box-plots show median (centre line); box limits are 25th to 75th percentile; whiskers represent min and max values. Statistical analysis was carried out using 1-way ANOVA, Tukey's multiple comparisons, CI = 95%, ****$p < 0.0001$; ***$p < 0.001$; **$p < 0.01$; *$p < 0.1$.

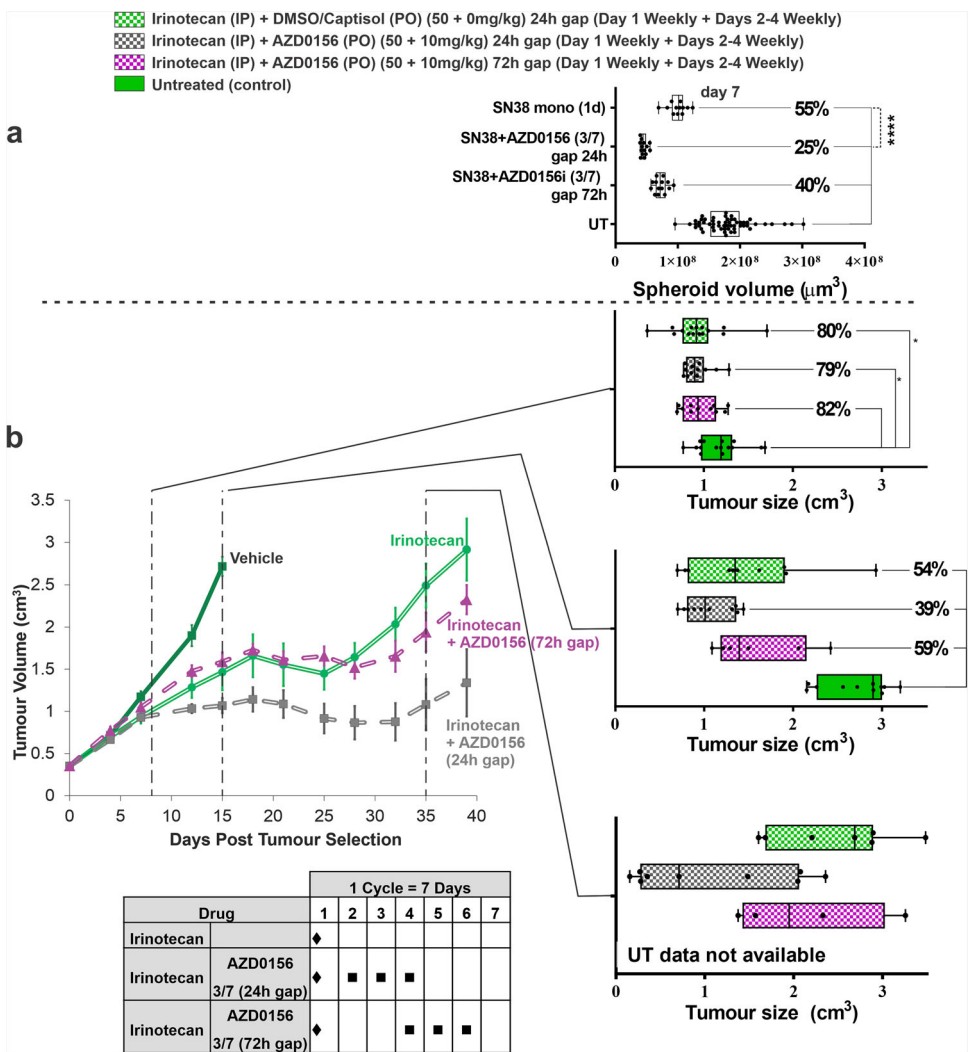

**Fig. 5 Comparison of SW620 tumour cells response to different schedules of topoisomerase-I inhibitor and AZD0156 in the microfluidic Ibidi chip versus mouse in vivo study.** SW620 tumour spheroids were treated with SN38 and two different gapped schedules with the ATM inhibitor in Ibidi microfluidic chip using 8-concentration profile over 24 h (Table 1), spheroids volumes were evaluated at day 7 of post-treatment. N ≥ 5 per condition from two independent experiments. Box-plots show median (centre line); box limits are 25th to 75th percentile; whiskers represent min and max values (**a**); SW620 tumour xenografts were subcutaneously implanted in mice (N = 15 per group) and subjected to the same treatment schedules as the spheroids in Ibidi chip. Tumour volume was measured regularly over 40 days. The graph represents mean ± SE. Tumour volumes at days 7, 15 and 35 were plotted as box-and-whisker graphs for easier comparison with in vitro microfluidic data (median—centre line; box limits—25th to 75th percentile; whiskers—min and max values) (**b**). Statistical analysis was carried out using 1-way ANOVA, Tukey's multiple comparisons, CI = 95%, ****$p < 0.0001$; ***$p < 0.001$; **$p < 0.01$; *$p < 0.1$.

technologies and pump-driven or pump-free (hydrostatic pressure) microfluidic systems. Most of these platforms, however, are only able to deliver constant flow rates or drug concentrations one at a time, on either a 2D or 3D cell model (from monolayers to complex cultures and organoids)[15,20,21]. Only one recently published novel design[22] mimics a compound's PK profile in a tubing-free, hydrostatic pressure setup. In this system, the proposed chip is designed to

deliver the typical drug plasma concentration decrease over time by changing the tilting angle that influences the flow rate. The pump-driven platform we have developed and described here, is able to deliver constant or customised flow rates, mimicking various physiological scenarios. While microfluidic platforms including vascular cells/endothelium equivalents require variable flow rates for cell seeding and capillary development[23], our setup focused on a

constant flow rate, consistent with the average tumour flow rates. Blood perfusion ranges between 8–100 mL/100 g × min in normal tissues and from a high of 100 mL/100 g × min to complete stasis in tumours[24]. We have opted for a 20 μL/min flow rate that is at the low end of those used in other in vitro microfluidic work (15–200 μL/min)[25–30]. The flow rate was kept relatively low to both reduce the stress on the spheroids, and to ensure access to the compounds and nutrients given that in our MF platform the spheroids are fed by diffusion through the Matrigel, rather than direct contact with the culture medium. Most prototype microfluidic devices are gas-permeable, fabricated from poly-dimethylsiloxane (PDMS) by soft lithography[31]. However, PDMS is well known as a small-molecule absorber that can, consequently, affect the concentrations delivered to the tumour model[32]. The platform we describe here has avoided the use of PDMS in contact with drug and is using an air +5% $CO_2$ mixture to push media into the flow line, with the generated air bubbles being retained in a nitrocellulose membrane bubble trap (Fig. 1).

In this study, we have been able to significantly improve on existing systems by combining a pump-driven microfluidic setup with an adapted single-channel Ibidi chip, allowing the flexibility to deliver a particular drug or drug combination for a certain time, corresponding to the physiological relevant PK profile. The fact that each Ibidi chip can handle up to eight tumour spheroids encapsulated in a Matrigel droplet, means the system is able to cope with the multiple variables associated with both biological complexity, while providing mechanical stability to the 3D culture under flow.

Our findings show that the response of SW620 cells to the treatments in plate static format are different to those obtained from the chip microfluidic setup. Viability of cells exposed to SN38 for 6 days in the plate is about 20% lower than in the microfluidic setup. Also, the combination with AZD0156 resulted in greater reduction in viability in the plate format (2.2-fold decrease) versus the Ibidi microfluidic chip (1.4-fold decrease). These results support the notion that non-physiological exposures to drugs in vitro may not deliver accurate response readout and more advanced in vitro systems, such as the platform we developed, should be considered for more translational drug assessment.

Another significant advantage of the microfluidic Ibidi chip over the static in vitro plate format is the feasibility to explore treatment scheduling. Understanding how different sequencing (continuous, intermittent, gapped schedules) of drugs impact the efficacy outcome can help prioritise the design of confirmatory in vivo studies, and thus limit the animal requirement as well as reducing time to generate these data to inform the clinical studies. Importantly, our proof-of-concept study with irinotecan and AZD0156 suggested that the results generated from this microfluidic platform are indeed translatable into in vivo setting.

We also demonstrated with biomarkers that the combination of SN38 and AZD0156 potentiates DNA double strand breaks and cell death induction and reduces proliferation in SW620 tumour spheroids (Fig. 4). We propose that the choice of combination treatment schedule could be further guided by changes in pharmacodynamic markers that can be readily evaluated in our chip-microfluidic system.

Further development of our platform is possible, allowing additional automation of steps and increased numbers of flow paths. For example, the OB1 MK3 pump can drive four flow paths, each able to deliver up to nine different concentrations of a drug's PK profile. Each flow path could then safely deliver treatment on two microfluidic chips, loaded by up to eight encapsulated spheroids each. Moreover, this microfluidic platform can be easily adapted to different microfluidic chips, using simple flow adaptors for various inlet/outlet types. For example,

by chaining tumour chips and normal tissues-on-chip in the current setup it may be possible to enable simultaneous assessment of the combination efficacy and toxicity, essentially therapeutic index on a chip, an essential component that needs to be understood to progress the development of any oncology drug. Both the flexibility and modularity of this PK profile-mimicking microfluidic platform, together with potential for up-scaling, automation and reasonable running costs, make this an important evolution in the development of the organ-on-chip concept.

## Methods

**Cell line**. Colorectal cell carcinoma cell line SW620 (CCL-227, ATCC) at passage 88 was maintained in Leibovitz's L-15 culture medium (L-15; 11415064, Gibco, ThermoFisher), supplemented with 10% foetal bovine serum (FBS; 16000044, Gibco, ThermoFisher), 1% Sodium Pyruvate (NaPyr, 100 mM; 11360070, Gibco, ThermoFisher) and 1% Glutamax (100×, 35050061, Gibco, ThermoFisher). This will be referred as L-15 complete media. All cultures were maintained in a humidified 5% $CO_2$ incubator at 37 °C. Cells on passages 90–98 were used to generate spheroids to be tested in static (plate format) and microfluidic setup.

**Spheroid generation and encapsulation**. Static and microfluidic setup were run simultaneously, using cells from the same batch. SW620 cells were thawed and proliferated for two passages and less than eight passages were used to generate spheroids. Cells in the flask were detached at 75% subconfluence, using TrypLE™ Express Enzyme (1X), no phenol red (12604013, Gibco, ThermoFisher). $1.5 \times 10^3$ cells in 200 μL cooled L-15 complete media with 1% Matrigel (phenol-red free, 356237, Corning), supplemented with 5% FBS only, were distributed in 96-round bottom well ultra-low attachment plates (650901, Greiner Bio-One). Following cell dispensing, cells were forcefully sedimented, using a cooled centrifuge (Sorvall Legend RT, ThermoScientific) at 300×g for 5′. Spheroids were allowed to bulge for 48 h and then distributed for static (plate format) and microfluidic setup testing. The approximate size of spheroids at day 0 were between 250–350 μM, and on day 1 (day of treatment) the size was between 290–370 μM and there was no statistical difference between those in the static or microfluidic set-up.

Static setup (plate format) used spheroids formed in 96-round bottom well ultra-low attachment plates (650 901, Greiner Bio-One) and no Matrigel encapsulation.

In the microfluidic setup, to avoid spheroid washout and mimic tumour microenvironment, spheroids were encapsulated in Matrigel droplets and transferred as hanging drops to the inner top side of the Ibidi chip channel (Supplementary Method 2). Ibidi chip channel was sealed using the corresponding polymer coverslip. The chip was incubated at 37 °C for 10 min to allow Matrigel gelation and full encapsulation, prior to filling with cell culture medium and connecting the chip to the flow path.

**Microfluidic setup**. To mimic in vivo anti-cancer drugs PK profile at a constant flow rate, an 8-point concentration scale for specific durations was used. Each drug was delivered at a specific concentration, for a specific amount of time, consistent with the average concentration for the same durations in vivo, on a 24-h step schedule.

L-15 complete media supplemented with 5% FBS only (to minimise protein binding) and 1% Penicillin-streptomycin was used for all experiments within microfluidic platform.

Medium flow was provided by a 4-channel OB1 MK3 microfluidic piezoelectric controller Elvesys, France). Liquid flow was generated by the airflow directed by OB1 MK3 controller over culture medium reservoirs. Flow rate was set/adjusted at the start of the experiments. A priming step using media with no drug for 20 min was used to allow the fluidic path to accommodate to media flow and the flow rate to stabilise. Airflow at OB1 MK3 input was a mixture of air and 5% $CO_2$ provided by a compact gas mixer (539200095, MM-Flex, Witt) fuelled by air and $CO_2$ wall lines (4 bar). $CO_2$ has been dissolved in the reservoir culture media with no requirement for permeable parts on the microfluidic chip. No PDMS parts were used, to prevent compound adsorption in any part of the setup. Airflow dispensed by each of the OB1 MK3 channels was split by a 10-port custom made low-pressure air manifold, to reach all reservoirs simultaneously. Liquid flow from reservoirs was directed in the flow path once at a time, corresponding to each of the eight points on the mimicked PK profile, due to the software-controlled piezoelectric switch in the MUX distributor.

**The flow path included:**.

Culture medium reservoirs:

50 mL conical super clear tubes, PEN0777696, Perkin-Elmer) with 50 mL reservoir connection cap (two port, standard 1/4″/28 fittings, Darwin Microfluidics, France);

100 mL and 250 mL Pyrex bottles (139550, Corning, ThermoFisher) with GL45 thread cap (two port, standard 1/4″-28 fittings, Darwin Microfluidics, France);

10/11 port MUX distributor valve (Elvesys, France)
Digital thermal flow sensors feedback (80 µL/min, Elveflow, France);
Bubble trap kit (97 µL, medium, Darwin Microfluidics, France) to avoid air bubbles in the chip,
1/16′ OD FEP tubing (GZ-06406-60, Cole-Parmer);
Fittings (ferrules, Luer connectors);
1-channel sticky bottom chip to handle hanging drop spheroids (0.8 mm channel height, Ibidi, Germany) with sealable bottom (polymer coverslip);
Flow resistors

A constant flow rate of 20 µL/min was self-adjusted by real-time pressure control using the flow sensors. Most parts of the microfluidic setup are reusable. Cleaning and sterilisation for microfluidic setup parts.

All polycarbonate (PC) parts were kept in 70% Ethanol overnight; prior experimental setup build-up and joining parts, all ethanol-cleaned parts were dried in a class-2 hood, in individual sterile Petri dishes;
Nitrocellulose membranes from bubble trap systems and Ibidi polymer film for sealing chip's bottom were exposed to UV on both sides for 10 min in a transilluminator;
All PEEK and glass pars were autoclaved at 115 °C for 15 min.

Each microfluidic setup included two chained Ibidi chips per flow line, to allow simultaneously assessment for viability (chip1) and biomarkers (immunofluorescence, chip2).

**Treatment setup.** Compounds used were SN38 (the active metabolite of the TOP1 inhibitor—irinotecan) and AZD0156 (a first-in-class oral selective inhibitor of ataxia telangiectasia mutated protein kinase (ATM), ATMi.

To mimic in vivo plasma PK profile for SN38 in monotherapy or combination with AZD0156, eight points (concentrations) from the in vivo PK profiles were considered, each of them with specific and synchronised timings (Supplementary Method 3). Each compound concentration/timing represented an average of the corresponding in vivo plasma values, measured by mass spectrometry (Supplementary Methods 4 and 5).

For static setup, compounds were used at the top concentration points used at the Cmax of the PK profile (4.5 nM for SN38 and 192 nM for ATMi). Compounds were dispensed in each well using an automated HP D300 digital dispenser (Tecan, UK).

For microfluidic setup, concentrations used in corresponding reservoirs were obtained by 2-step dilution (Supplementary Method 6). On the first step, DMSO stock solutions of 0.1 mM SN38 and 1 mM AZD0156 were used to generate an array of 5 × 200 µL of 1 µM SN38 and 19.2 µM AZD0156, respectively, using the HP D300 digital dispenser. On the second step, final concentrations in corresponding reservoirs were obtained by manual pipetting and thorough mixing in the flow reservoirs. Each reservoir was provided with a least 5 mL volume to spare. Final dilutions were obtained L-15 complete media supplemented with 5% FBS only and 1% Penicillin-streptomycin.

Reservoirs were mounted in the threaded connection caps and L-15 media was flushed for 5′ min at 20 µL/min from the L-15 untreated reservoir. OB1 MK3 and MUX unified software controller (Elvesys, France) allowed sequential control of the medium + compound distribution from each reservoir for the designated amount of time. Schedules were designed and saved for each treatment condition. Each setup and treatment condition were repeated at least three times.

**Viability assessment.** Spheroid viability assessment used a metabolic method (CellTiterGlo 3D, G9681, Promega) by measuring luminescence units expressed by individually recovered spheroids. Each chip bottom polymer film was cut using a scalpel and spheroids were individually collected and transferred in 25 µL media to the wells of a half area 96-well plate (1/2 AreaPlate-96, white, 6002290, Perkin-Elmer). CellTiterGlo vial was allowed to thaw (1 h, room temperature) and 25 µL reagent were added to each spheroid well. Plates were lidded with black sealing tape (Nunc, 731-0750, VWR) and placed on and orbital shaker for 40′. Luminescence levels were measured using a SpectraMax i3X (Molecular Devices, USA).

**Immunofluorescence and imaging.** For biomarker evaluation, at day 7, spheroids in each chip were washed in PBS for 5′, fixed in 4% paraformaldehyde solution (J19943.K2, Thermo Scientific) for 30′, permeabilized (0.3% Triton™ X-100 in PBS, T8787, Sigma-Aldrich) for 30′ and blocked (0.3% BSA in PBS) for 30′, using the OB1 MK3 controller to generate a flow rate of 80 µL/min. Then, triple labelling was performed overnight, in static conditions, using primary conjugated antibodies for γH2Ax (gamma H2AX [p Ser139] Antibody [Alexa Fluor 488], NB100-384AF488, Novus Bio/BioTechne), caspase-3 (CC3, Caspase-3 Antibody (31A1067) [Alexa Fluor 594], NB100-56708AF594, and KI67 (Anti-Ki67 antibody [EPR3610] (Alexa Fluor 647], ab196907, Abcam) at 1:500 dilution each. Nuclear counterstaining was performed by Hoechst 33342 at 1:5000 dilution (H3570, ThermoFisher).

**Image processing.** Spheroid images were taken for each experimental endpoint (day 7) using a Leica DMI4000B microscope (Leica Microsystems) with brightfield (BF), phase-contrast and fluorescence filters. Each flow line used two chips, one for viability assays and second for immunofluorescence. Both chips were imaged in BF prior spheroid recovery. Hardware embedded microscope scale has been translated in Fiji ImageJ to evaluate spheroid area/volume. The average size for SW620 spheroids was evaluated by measuring spheroid projection area in bright-field images, based on a macro script designed in an open-source imaging software (Fiji ImageJ) (Supplementary Method 1). Spheroid volume was evaluated following radius deduction from the measured area.

Immunofluorescence Matrigel-embedded and stained spheroids, labelled for γ-H2Ax, CC3 and Ki67, were removed from the Ibidi chip, following polymer film removal, using a wide bore 200 µL tip, to avoid spheroid squeezing. Spheroids were transferred into 96-well round bottom plates, in 50 µL PBS, then images were generated on a CV7000 confocal microscope. Images were transferred and analysed in Fiji ImageJ, using macros designed and tailored for each fluorescence channel and biomarker respectively. Macros and image analysis protocol for individual biomarkers are described in Supplementary Fig. 2 and Supplementary Method 7.

### In vivo studies

*In vivo PK profile determination.* The pharmacokinetic analysis of AZD0156 was performed in Swiss athymic nu/nu male mice (n = 3 per timepoint) dosed orally at 5 mg/kg dose and PK samples taken at 0.5, 2, 6 and 24 h via venipuncture of the tail vein. PK profile of irinotecan was investigated in CB-17 SCID female mice dosed intraperitonially at 50 mg/kg. The samples were collected at 5, 15, 40 min, and 1, 2, 4, 6, 8 and 24 h.

Whole blood was mixed 1:5 with PBS and centrifuged at 1500×g for 3 min at 4 °C, and the plasma was extracted and frozen at –80 °C.

The measured mouse plasma concentrations of AZD0156 and irinotecan with its active metabolite were fitted to obtain PK parameters using population-based PK model (described in Supplementary Method 8). Using this PK model, a time course of free plasma concentrations of both drugs was simulated (Table 1).

*Plasma bioanalysis.* Each sample (25 mL) was prepared using an appropriate dilution factor, and compared against an 11-point standard calibration curve (0.05–500 nM) prepared in DMSO and spiked into blank plasma. Acidified acetonitrile (100 mL) was added with the internal standard, followed by centrifugation at 3000 rpm for 10 min. Supernatant (60 mL) was then removed to a clean plate and dried under nitrogen. Samples were reconstituted in 150 µL water:acetonitrile, formic acid (90:10, 0.1%) and analysed via UPLC-MS/MS. Mass spectrometer and UPLC parameters are detailed in Supplementary Method 5.

*Treatment efficacy study.* Male immunocompromised Hsd:Athymic Nude-Foxn1nu mice (Envigo) were used for xenograft tumour implantation. SW620 (ATCC® CCL-227™) xenograft was established by implanting 100 µL of a cell suspension (1 × 10^6 cells in 50% Matrigel (BD Bioscience)) subcutaneously into the dorsal left flank of the animals. Animals were randomised into groups of 9–15 when tumours reached a volume of ~0.30 cm³ and 4 weekly cycles of treatment commenced.

AZD0156 was formulated in a 10% DMSO/90% Captisol (30%w/v) solution (Cydex Pharmaceuticals) and orally dosed. Irinotecan was formulated in a 7.5% DMSO/92.5% water for injection solution and administered once a week via the intra peritoneal (IP) route (50 mg/kg). In the combination group animals were treated with irinotecan (50 mg/kg IP) followed 24 or 72 h by three consecutive daily oral doses of AZD0156 (10 mg/kg)). This was repeated for four consecutive weeks. Control animals were treated IP once weekly with 0.85% Physiological saline and PO with 10% DMSO/90% Captisol (30% w/v) once daily for 3 days per week 24 h after the IP weekly dose.

Tumours were measured twice weekly (length × width) by bilateral Vernier calliper measurements and tumour volume calculated using Mousetrap software. Tumour growth inhibition from start of treatment was assessed by comparison of the mean change in tumour volume for the control and treated groups using the Mousetrap application and represented as TGI.

All in vivo studies complied with all relevant ethical regulations for animal testing and research, followed AstraZeneca's global bioethics policy and received ethical approval from the AstraZeneca ethical committee. All studies were conducted in the UK in accordance with UK Home Office legislation, the Animal Scientific Procedures Act 1986 and under Home Office project licence 40/8894.

*Statistics and reproducibility.* All experiments are performed at least in triplicate or as indicated. One-way ANOVA with Tukey's multi-comparison post-test was performed on data using Prism 8 (GraphPad Software, Inc., La Jolla, CA).

**Reporting summary.** Further information on research design is available in the Nature Research Reporting Summary linked to this article.

### Data availability

The source data for the graphs and charts in the figures is available as Supplementary Data 1 and any remaining information can be obtained from the corresponding author upon reasonable request.

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

## Author contributions

T.P. and L.O.OC wrote the manuscript with input from M.A., A.L., S.C., and M.OC. T.P. designed and set up the microphysiological system (MPS), generated all the experiments with tumour on the chip and analysed the data. M.OC. and E.C. contributed to the design of the experiments carried out on the MPS and E.C. supervised the in vivo studies. G.H. was involved in the experimental design and carried out the in vivo studies. A.S. was responsible for analysis of plasma samples from pharmacokinetic studies. V.P.R. performed modelling of the pharmacokinetic data. All authors reviewed the manuscript. L.O.OC and M.A. contributed to the experimental design and interpretation of data generated on the MPS and both were responsible for conception and oversight of the project.

## Competing interests

The authors declare no competing interests.
