## [Peer Review File · Communications Biology]

Reviewers' Comments:

Reviewer #1:

Remarks to the Author:

The authors developed a platform for the delivery of specific drug concentrations at constant flow rate to assess the response of tumour spheroids on different PK profiles. The article is well written and of interest. However some things needs to be clarified since it is now not clear if the differences are caused by the microfluidic environment and/or encapsulation in matrigel.

Major comments

- On page 5 line 116 it is mentioned that the Matrigel encapsulated spheroids can resists the medium flow for longer term experiments when compared to nude ones. What kind of shear stresses are expected on the spheroids, and what was the corresponding flow rate. The dimensions of the chip and the embedding in gel will influence this and it is good for the reader to understand how you can tweak this (and also what the maximum flow rate can be without damaging the spheroids in both situations).
- The authors normalized the tumour growth and this is understandable. However, can the authors also tell something about the sizes on day 1 for the different cultures (3D, 2D, Matrigel?) Are their differences in this and if so, does this influence the result?
- In the discussion it is mentioned that the spheroids are fed by diffusion. This all depends on the thickness of the Matrigel layer, the thickness of the spheroid, pore sizes, molecule size. Can the authors elaborate on this? So what is a typical time scale for the diffusion and how does this change the results with respect to the static culture/culture without hydrogel? Is the Matrix layer always the same size or does this not really influence the concentration profile?
- It is not really clear if the static setup (plate format) also uses Matrigel. If not, is this comparison really comparing the differences between microfluidic setup and the plate or does it also include the effect of using a Matrigel. To test this, a plate format with Matrigel should be tested as well.
- Flow rates were adjusted in the platform. Did the authors take the dead volumes into account (maybe it is good to mention this in the text for people who are not familiar with such a setup). Furthermore what is the RC time? After adjusting the flow, this probably will not immediately directly respond to the flowrate (has to do with capacitances and resistors in the system). Did you take this into account or is it neglectable?

Minor comments

- Page 2, line 61: different font size
- Page 4, line 106. First time abbreviation LC MS
- What is the idea of the text below figure 1. This is informative, but not part of the legend.

Some state of the art literature is missing. Please include:

Y. A. Guerrero et al. The AAPS journal, 2020, DOI: 10.1208/s12248-020-0430-y

J. Komen et al. Lab on a chip, 2020, DOI: 10.1039/d0lc00419g

D. B. Chou et al. Nature biomedical engineering, 2020, DOI: 10.1038/s41551-019-0495-z

Reviewer #2:

None

Reviewer #3:

Remarks to the Author:

This manuscript proposed a new tumor-on-chip microfluidic platform to assess drug pharmacokinetics and treatment response in vitro. The results are promising; however, the writing of the manuscript should be improved. There are some also concerns about the figures should be addressed as well.

1. This manuscript proposed a new tumor-on-chip microfluidic platform; however, the authors never showed an image and/or design layout of the chip with details. Figure 1 only shows a small picture of

the chip, which is not sufficient to for audiences to understand the chip design.

2. The authors first mentioned Figure 2b on page 4, line 112, before they mentioned Figure 2a. The order of these subfigures should be updated.

3. The authors should calculate the p values between key groups in Figure 2c,d to show whether there are significant changes.

4. The figure legends were not written in a concise manner, thus should be revised.

2nd June, 2021

Response to referees: COMMSBIO-20-3593-T: A novel tumour-on-chip microfluidic platform advances assessment of drug pharmacokinetics and treatment response

Please find below our point-by-point answers to the two reviewers. To facilitate the review process, we have included the original comments in italic and quote marks, and our answers below.

Reviewer 1

1. *“On page 5 line 116 it is mentioned that the Matrigel encapsulated spheroids can resist the medium flow for longer term experiments when compared to nude ones. What kind of shear stresses are expected on the spheroids, and what was the corresponding flow rate. The dimensions of the chip and the embedding in gel will influence this and it is good for the reader to understand how you can tweak this (and also what the maximum flow rate can be without damaging the spheroids in both situations).”*

Our response: Individual spheroids were encapsulated in 50% Matrigel hemispherical droplets of approximately 1mm in diameter. The chip is a rectangular chamber 0.8mm height x 5mm width x 50mm length. Using the Darwin microfluidic website, we calculated the shear stress in the rectangular microfluidic chip (Ibidi μ -Slide I Luer) for 20 μ L/min flow rate to be 6.25×10^{-3} dyne/cm².

It is important to mention that the shear stress applies to the Matrigel hemispherical droplet and not directly on the encapsulated spheroid. As long as the flow rate does not exceed 70 μ L/min for 3 days or longer, the Matrigel encapsulated spheroids can withstand the culture media flow. For long term experiments (>2 weeks), we observed that 20-30% of the Matrigel droplets were being removed at a flow rate of 50 μ L/min. At a flow rate of 20 μ L/min, none of the Matrigel droplets containing embedded spheroids were removed from the chip.

The Matrigel droplet is a hanging drop hemisphere on the inner top side of the microfluidic chip. It's adherence to the surface could be augmented for potential use with higher flow rates in the chip. However, higher flow rates may not be physiologically relevant for this current model in the absence of tumour model microvascularization.

As requested, we have addressed the concerns that were highlighted and added more details into the manuscript (Page 3/ line 113) “The Matrigel encapsulated spheroids in the Ibidi chip can resist the medium flow rate of 20 μ L/min for up to 14 days. For flow rates greater than 20 μ L/min but not exceeding 70 μ L/min, the encapsulated spheroids were able to withstand the culture media flow for at least 3 days.”

2. *“The authors normalized the tumour growth and this is understandable. However, can the authors also tell something about the sizes on day 1 for the different cultures (3D, 2D, Matrigel?) Are their differences in this and if so, does this influence the result?”*

Our response: To generate SW620 spheroids, we seeded an initial 1500 cells per each spheroid. After 48h, the spheroids were transferred to 96-well plates (static experiment) or Ibidi chip. The size of the 3D spheroids at the beginning of the treatment were similar for both static and microfluidic experiments. The average diameter of the spheroid on day 0 (day of transfer) was 250-350 μ M and on day 1 (day of treatment) the size was between 290-370 μ M. To make this clear to the reader the following sentence has been added to the methods (Page 11/ line 330) “The approximate size of spheroids at day 0 were the e between 250-350 μ M and on day 1 (day of treatment) the size was between 290-370 μ M and there was no statistical difference between those in the static or microfluidic set-up”. Experiments on 2D cultures were not included in this work.

3. *“In the discussion it is mentioned that the spheroids are fed by diffusion. This all depends on the thickness of the Matrigel layer, the thickness of the spheroid, pore sizes, molecule size. Can the authors elaborate on this? So what is a typical time scale for the diffusion and how does this change the results with respect to the static culture/culture without hydrogel? Is the Matrix layer always the same size or does this not really influence the concentration profile?”*

Our response: We used 50% Matrigel in our experiments with average pore size about 2 μ m. The spheroids were generated from the same number of cells and were 250-350 μ m in diameter when they were individually encapsulated in Matrigel hemispherical droplets measuring approximately 1mm in diameter. The exact position of the spheroid in the droplet is random however, so the layer of Matrigel around each spheroid is variable (see Figure 1).

Figure 1: Spheroid encapsulation in Matrigel in Ibidi chip

However, we do not think that the Matrigel layer thickness variability could influence the response to the treatment as the pores are large enough to ensure small molecule (SN38, ATM inhibitor) diffusion. In the manuscript, we show data that were generated from 6 technical replicates for each experimental condition

$$D = \frac{kT}{6\pi\eta R}$$

Based on Stokes-Einstein equation $D = \frac{kT}{6\pi\eta R}$, the diffusion coefficient in our experiments ranges from $\sim 100\mu\text{m}^2/\text{sec}$ to $\sim 1000\mu\text{m}^2/\text{sec}$. In the microfluidic setup, each drug concentration was flowing through the chip for a minimum 1h. The drugs we tested should diffuse through the Matrigel in seconds and not be affected by the spheroid position in the hemispherical Matrigel droplet. Our assumption is therefore that the response to the treatment was not influenced by the encapsulation procedure; and the reproducibility of the data would support this. The text has been modified (Page 8/ line 258), however, to make this clearer. “The flow rate was kept relatively low to both reduce the stress on the spheroids and to ensure access to the compounds and nutrients given that in our MF platform the spheroids are fed by diffusion through the Matrigel rather than direct contact with the culture medium.”

4. *“It is not really clear if the static setup (plate format) also uses Matrigel. If not, is this comparison really comparing the differences between microfluidic setup and the plate or does it also include the effect of using a Matrigel. To test this, a plate format with Matrigel should be tested as well.”*

Our response: Our static plate experiments did not include Matrigel and the spheroids were directly incubated with the drugs in the culture medium. We expect, however, that the nude and Matrigel encapsulated spheroids receive comparable exposure (please refer to the previous answer regarding the drug diffusion). The primary purpose of the Matrigel encapsulation here was to provide a mechanical support for the spheroids in the flow path without affecting the drug penetration. We therefore believe that the experiment testing the treatment of the spheroids in the plate with Matrigel would be redundant.

However, to provide some further insight onto the effect of Matrigel on spheroids growth in both plate and the chip set up, we provide here some additional data from our initial experiments comparing the growth rate of untreated “nude spheroids” (no Matrigel, plate format) versus Matrigel encapsulated spheroids in Ibidi chip with no flow or $20\mu\text{L}/\text{min}$ flow rate. After 6 days, we observed 4.2 fold increase in the “nude spheroids” volume, compared to 3.44 for Matrigel-encapsulated spheroids in Ibidi chip with flow, and 3.15 fold for Matrigel-encapsulated spheroids in Ibidi chip with no flow.

Importantly, the microscopic images (Figure 2) showed significantly reduced necrotic core in the microfluidic setup under the flow, which suggests more optimal conditions for tumour spheroids cultivation.

Figure 2: Microscopic images and size of untreated SW620 spheroids grown for 6 days in the plate and Ibidi chip

5. “Flow rates were adjusted in the platform. Did the authors take the dead volumes into account (maybe it is good to mention this in the text for people who are not familiar with such a setup). Furthermore what is the RC time? After adjusting the flow, this probably will not immediately directly respond to the flowrate (has to do with capacitances and resistors in the system). Did you take this into account or is it neglectable?”

Our response: Flow rate was adjusted only at the start of the experiments. A priming step using media with no drug for 20 min was used to allow the fluidic path to accommodate to media flow and the flow rate to stabilize. The piezo pump controller is very accurate and the response to any flow variation detected by the flow sensor installed on the flow path is adjusted within seconds. All fittings, connectors, bubble traps, tubing length were identical for all conditions, thus generating similar dead volumes for all the experiments. Generally, the “front wave” required approximately 60 sec to reach the waste port.

As suggested, we have added more information around this to the manuscript (Page 12/ line 353) “Flow rate was set/adjusted at the start of the experiments. A priming step using media with no drug for 20 min was used to allow the fluidic path to accommodate to media flow and the flow rate to stabilise.”.

6. “Some state of the art literature is missing. Please include:

Y. A. Guerrero et al. The AAPS journal, 2020, DOI: 10.1208/s12248-020-0430-y

J. Komen et al. Lab on a chip, 2020, DOI: 10.1039/d0lc00419g

D. B. Chou et al. Nature biomedical engineering, 2020, DOI: 10.1038/s41551-019-0495-z”

Our response: We agree these are very relevant references and we have now included them in the manuscript.

Reviewer 3

1. “This manuscript proposed a new tumor-on-chip microfluidic platform; however, the authors never showed an image and/or design layout of the chip with details. Figure 1 only shows a small picture of the chip, which is not sufficient to for audiences to understand the chip design.”

Our response: Thank you for highlighting this. We agree this would be helpful and have now provided real images of the Ibidi chip with encapsulated spheroids in Figure 1b. The chip is a commercially available Ibidi Luer chip and the set up to our microfluidic system is described in details in the Methods.

Figure 1b in the manuscript:

2. *“The authors first mentioned Figure 2b on page 4, line 112, before they mentioned Figure 2a. The order of these subfigures should be updated.”*

Our response: We have rectified this error.

3. *“The authors should calculate the p values between key groups in Figure 2c,d to show whether there are significant changes.”*

Our response: We have now included p-values as requested.

4. *“The figure legends were not written in a concise manner, thus should be revised.”*

Our response: Thank you for your feedback, the figure legends have been revised and we feel they are now more concise.

REVIEWERS' COMMENTS:

Reviewer #3 (Remarks to the Author):

The authors have addressed the reviewers' comments.